# Changes in the Brain Metabolism Associated with Central Post-Stroke Pain in Hemorrhagic Pontine Stroke: An ^18^F-FDG-PET Study of the Brain

**DOI:** 10.3390/brainsci12070837

**Published:** 2022-06-27

**Authors:** Soo-jin Choi, Na-young Kim, Jun-yup Kim, Young-sil An, Yong-wook Kim

**Affiliations:** 1Department and Research Institute of Rehabilitation Medicine, Yonsei University College of Medicine, 50-1 Yonsei-ro, Seodaemun-gu, Seoul 03722, Korea; sjc8070@naver.com (S.-j.C.); kny8452@yuhs.ac (N.-y.K.); 2Department of Medicine, Graduate School, Yonsei University College of Medicine, Seoul 03722, Korea; 3Department of Rehabilitation Medicine, Bucheon St. Mary’s Hospital, College of Medicine, The Catholic University of Korea, Bucheon-si 14647, Korea; 4Department of Rehabilitation Medicine, Yonsei University Yongin Severance Hospital, Yongin 16995, Korea; 5Department of Physical Medicine and Rehabilitation, Hanyang University Medical Center, Seoul 04763, Korea; futurer22c@gmail.com; 6Department of Nuclear Medicine and Molecular Imaging, Ajou University School of Medicine, Suwon 16499, Korea; aysays77@naver.com

**Keywords:** pain, pons, stroke, cerebral hemorrhage, cerebral cortex, cerebellum

## Abstract

Central post-stroke pain (CPSP) is an intractable neuropathic pain that can occur following central nervous system injuries. Spino-thalamo-cortical pathway damage contributes to CPSP development. However, brain regions involved in CPSP are unknown and previous studies were limited to supratentorial strokes with cortical lesion involvement. We analyzed the brain metabolism changes associated with CPSP following pontine hemorrhage. Thirty-two patients with isolated pontine hemorrhage were examined; 14 had CPSP, while 18 did not. Brain glucose metabolism was evaluated using ^18^F-fluorodeoxyglucose-positron emission tomography images. Additionally, regions revealing metabolic correlation with CPSP severity were analyzed. Patients with CPSP showed changes in the brain metabolism in the cerebral cortices and cerebellum. Compared with the control group, the CPSP group showed significant hypometabolism in the contralesional rostral anterior cingulum and ipsilesional primary motor cortex (*P_uncorrected_* < 0.001). However, increased brain metabolism was observed in the ipsilesional cerebellum (VI) and contralesional cerebellum (lobule VIIB) (*P_uncorrected_* < 0.001). Moreover, increased pain intensity correlated with decreased metabolism in the ipsilesional supplementary motor area and contralesional angular gyrus. This study emphasizes the role of the many different areas of the cortex that are involved in affective and cognitive processing in the development of CPSP.

## 1. Introduction

Central post-stroke pain (CPSP) is a neuropathic pain syndrome caused by a cerebrovascular lesion at any level of the central nervous system [1,2]. It is characterized by burning, aching, pricking, and annoying pain and signs of hypersensitivity in the body part topographically corresponding to the location of the lesion of the central nervous system, usually contralateral to the side of the stroke lesion [3]. CPSP is often intractable and can interfere with the quality of life, causing long-term discomfort. CPSP has been shown to be associated with not only thalamic lesions but also extra-thalamic lesions [3]. It had been mistermed as thalamic pain; however, any lesion, regardless of the level of injury, affecting the spino-thalamo-cortical pathway and its cortical projection, including the brainstem, thalamus, and cerebral cortex, can interfere with pain sensation and act as a crucial factor in the development of CPSP [3,4].

While multiple pathways transmit and modulate pain information, there does not appear to be a single “pain cortex” associated with the sensation of pain [5]. A network model of pain, which is an emerging concept, explains the sensation of pain through an integrated brain network [4]. The interactions between cortical and subcortical processing areas across the brain appear to constitute a complicated structural network for pain perception [4]. However, the cerebral cortical areas involved in pain processing are not yet completely elucidated.

Since CPSP usually occurs in supratentorial lesions, many previous neuroimaging studies are limited to supratentorial strokes with lesions involving the cerebral cortices and deep gray or white matter [6]. However, central pain can also develop following brainstem strokes, including pontine strokes [7]. Pontine involvement is most common among the brainstem hemorrhagic events. Unlike the ischemic strokes usually affecting the ventral part and causing motor dysfunction, pontine hemorrhagic strokes involve the dorsal pons and the ascending sensory fibers, which, as a result, can cause CPSP as well as sensory dysfunction [7]. Therefore, unlike previous studies, the present study focused on patients with pontine hemorrhage without cortical involvement who developed CPSP. Through this study, we aimed to evaluate the functionally related regions for processing pain and their involvement in CPSP by analyzing the metabolic changes of glucose on ^18^F-fluorodeoxyglucose-positron emission tomography (^18^F-FDG-PET) images in patients with pontine hemorrhage.

## 2. Materials and Methods

### 2.1. Participants

We collected medical records of all adult patients who had experienced the first episode of pure pontine hemorrhage and were admitted to a tertiary inpatient rehabilitation hospital between January 2010 and December 2019. All the patients underwent physical and neurological examinations and detailed history taking within 24 h following admission, along with pain evaluation. Patients were only included in the study if they had undergone ^18^F-FDG-PET studies in the hospital following the onset of a stroke. Records were obtained using the electronic clinical data retrieval system. The inclusion criteria were (1) first episode of stroke, (2) solitary hemorrhage in the pons confirmed by brain computed tomography (CT) or magnetic resonance imaging (MRI), (3) age 20 years or above, and (4) no severe cognitive impairment, with a Mini-Mental State Examination (MMSE) score of 23 points or higher. Participants with evidence of old cerebral strokes were excluded if the lesions were larger than 3 mm in diameter on MRI [8].

The initial stroke severity was evaluated according to the National Institutes of Health Stroke Scale (NIHSS). The Fugl–Meyer (FM) assessment motor and sensory subscales were used to measure sensorimotor impairments. The severity of depressed emotional status was assessed using the Geriatric Depression Scale (GDS). Lesion volumes were measured from CT scans using the ABC/2 formula [9]. Pain intensity was rated using the numeric rating scale (NRS) at initial admission. It was measured as the average pain intensity over the last 48 h, with a score of 0 indicating “no pain” and 10 indicating “worst intolerable pain.” Medications that could interfere with the sensation of pain and brain metabolism, such as antidepressants, anticonvulsants, anti-anxiety drugs, analgesics, muscle relaxants, and antipsychotics, were quantified using the Medication Quantification Scale (MQS) [10].

Pain and sensory abnormalities were considered as CPSP if they had newly developed following stroke and in the body parts corresponding to the brain territory of the stroke [1,3]. Other causes of pain, such as hemiplegic musculoskeletal shoulder pain, peripheral entrapment neuropathy, radiculopathy, and painful spasticity, were excluded based on signs and symptoms noted on physical examination, and on ultrasonography and electromyography [11,12,13]. Complex regional pain syndromes were excluded using the Budapest criteria [14].

The study was approved by the Institutional Review Board of Yonsei University Health System, Severance Hospital, and was conducted in accordance with the principles outlined in the 1964 Declaration of Helsinki and its later amendments. The requirement for informed consent was waived owing to the retrospective nature of the study.

### 2.2. Statistical Analyses of Clinical Data

To compare the baseline demographics and clinical information between patients with and without CPSP, data were analyzed using SPSS Statistics software, Version 25.0 (IBM, Armonk, NY, USA). The Shapiro–Wilk test was used to check the normality of continuous variables. Depending on the distribution nature of the data, an independent *t*-test was used as a parametric test, and the Mann–Whitney U test was used as a nonparametric test for continuous variables (age at diagnosis, onset duration, volume of the lesion, NIHSS, FM motor, FM sensory, MMSE, GDS, MQS, and NRS). Fisher’s exact test was used for categorical variables (sex and lesion location). Correlations between the NRS and all the continuous variables were analyzed using Spearman rank correlation coefficient to assess potential confounding factors. A two-tailed *p* < 0.05 was considered statistically significant in all statistical tests on demographic and clinical data.

### 2.3. Acquisition of Brain ^18^F-FDG-PET Images

^18^F-FDG PET/CT brain scans were obtained using a GE Discovery 600 PET/CT scanner (GE Medical Systems, Milwaukee, WI, USA) for 15 min. Adverse effects were monitored for 30 min following the injection. First, an initial low-dose CT scan was performed, and subsequently, a three-dimensional PET emission scan was performed. The transverse and axial resolutions were 4.8 mm full width at half-maximum, attenuation-corrected emission data reconstructed in a 128 × 128 × 35 matrix, with a pixel size of 1.95 mm × 1.95 mm × 4.25 mm.

### 2.4. Analyses of Brain ^18^F-FDG-PET Images

PET images were processed and analyzed using Statistical Parametric Mapping (SPM) software version 12 (Statistical Parametric Mapping 12, Wellcome Centre for Human Neuroimaging, London, UK; http://www.fil.ion.ucl.ac.uk/spm/, accessed on 25 December 2021) of MATLAB R2018a software (MathWorks, Natick, MA, USA). Since the sensory symptoms of the CPSP are generally contralateral to the side of the stroke lesion [3], data sets of patients with right side pain were flipped to relocate the lesions to the right side of the brain. The ^18^F-FDG PET/CT brain images were normalized using the standard PET template of the Montreal Neurological Institute to make them comparable between the participants. Then, the images were smoothed by a three-dimensional filter, a Gaussian kernel with full width at half maximum of 8 mm × 8 mm × 8 mm [15,16]. To remove global nuisance effects and to improve the sensitivity, the intensity values for each scan were normalized by means of proportional scaling in SPM [17,18].

The resulting PET images were compared between patients with and without CPSP using the voxel-wise two-sample *t*-test implemented in SPM for brain glucose metabolism investigation. An uncorrected threshold of two-tailed *p* < 0.001 was used to determine statistical significance, with a cluster size threshold of 10 voxels. In addition, multiple regression was performed to identify the regions that were associated with the pain intensity in the CPSP group, controlling for age as a nuisance variable [19]. The anatomical locations of the regions were labeled using an Automated Anatomical Labeling program (https://www.gin.cnrs.fr/AAL/, accessed on 25 December 2021) [20].

Voxel-wise corrections for multiple comparisons implemented in neuroimaging packages, such as SPM, have been shown to be conservative, repeatedly [21,22]. Owing to the limited variability of the lesions, we expected to find small effects within brain cortices. Therefore, we decided to identify changes at a statistical voxel-wise uncorrected threshold of two-tailed *p* < 0.001. To reduce false-positive findings, a cluster size threshold of 10 continuous voxels was required for significance, and we later replicated the results with a regions-of-interest approach using volumetric measurements.

## 3. Results

As a result, a total of 32 patients with unilateral pontine hemorrhage meeting the inclusion criteria were identified. Among the 32 patients, 14 with contralesional pain were assigned to the CPSP group, and the remaining 18 without CPSP were assigned to the control group (Figure 1). There were no statistically significant differences between the CPSP and control groups in terms of age, sex, duration since onset of stroke, laterality of the lesion, lesion volume, and NIHSS, FM motor/sensory, MMSE, GDS, and MQS scores (Table 1). The location and size of pontine hemorrhages and the corresponding location and pain characteristics for each patient with CPSP are depicted in Table 2. Among the continuous variables, none showed correlation with the NRS scores. Overlays of the lesion distributions for all participants of the two groups are presented in Figure 2.

There were significant differences in metabolism between the two groups. Compared with the control group, the CPSP group showed significant hypometabolism in the contralesional rostral anterior cingulate cortex (ACC) and ipsilesional primary motor cortex (M1) (Figure 3, Table 2). However, increased brain metabolism was observed in the ipsilesional cerebellum (lobule VI) and contralesional cerebellum (lobule VIIB), which are in the superior posterior lobe of the cerebellum (Figure 4, Table 3). 

Multiple regression analysis revealed supratentorial cortices to be associated with the pain intensity expressed using the NRS scores. Results showed that decreased metabolism in the ipsilesional supplementary motor area and contralesional angular gyrus was correlated with increased pain intensity, and no region showed positive correlations (Figure 5, Table 4). None of the other variables showed a significant correlation.

## 4. Discussion

Several studies using brain ^18^F-FDG-PET images have investigated the neural correlates for the development of CPSP in supratentorial stroke. However, this is the first study to assess the metabolic changes in the brain related to CPSP following isolated brainstem stroke.

Although the CPSP and control groups did not show any significant difference in the baseline clinical parameters, differences in metabolism were found in multiple regions of the cerebral cortices and cerebellum. Additionally, altered brain metabolism in the ipsilesional supplementary motor area and contralesional angular gyrus were correlated with the pain intensity.

The supplementary motor area and angular gyrus were the only regions that appeared to be relevant to the pain sensitivity. Many aspects of the supplementary motor cortex remain questionable; however, its relation to emotion, affective functions, and cognitive control and behavior processing are known [23]. Moreover, its functional connections with the limbic system and primary motor cortex have been proven to play a role in negative emotions [24]. A study of rheumatoid arthritis patients using functional MRI (fMRI) reported increased connectivity, predominately for the supplementary motor cortex, cingulate cortex, and bilateral sensorimotor cortex, suggesting their involvement in pain processing [25]. The angular gyrus is believed to play an integrative role; multisensory inputs are integrated in the angular gyrus, and interactions with different subsystems such as memory, attention, and concepts are performed to ultimately comprehend events, thus acting as an attentional subsystem [26]. The angular gyrus thus reflects the ability to integrate aspects of information, especially including sensory information and internal mental representations [26].

Another cerebral cortical region, the anterior cingulate cortex (ACC) along with supplementary motor area, and angular gyrus, was found to be associated with CPSP in this study. The ACC has been reported to play a critical role in the process of emotional and cognitive tasks and pain perception, playing a role in the negative affective responses to pain sensation [27,28]. The ACC is also known to be a part of the ascending pain-related and descending inhibitory pain modulatory pathways [29,30]. Our results suggest that these cortical regions might presumably be involved in the processing of sensory information regarding the affective dimension of pain.

Many chronic pain syndromes have shown altered excitability of the primary motor cortex (M1) [31], and it is the major target for neuro-modulation by brain stimulation therapy to improve the symptoms in patients with intractable neuropathic pain [32,33]. The repetitive transcranial brain/magnetic stimulation of the motor cortex was shown to increase cerebral blood flow in many cortical and subcortical regions, including the ACC [32]. In previous studies with fMRI, the M1 showed markedly increased activation in relation to chronic pain or nociceptive stimuli, which provides evidence for a spinothalamic-tract-related input to the M1 [34,35]. However, until now, the correlation between the M1 and pain sensation has not been clearly elucidated, and our study might provide additional supportive evidence.

The cerebellum is engaged in a number of integrative functions, such as cognitive and affective functions, motor control, and somatosensory processing, including nociception [36,37]. It receives afferent nociceptive stimuli, and PET studies and neuroimaging studies using fMRI suggest that nociceptive-related activation is processed in the cerebellum [38,39]. Moreover, cerebellar cognitive affective syndrome is known to result from injury to the posterior cerebellar lobe [40]. Cognitive processing in areas of the cerebellum could be related to nociceptive processing and pain perception.

There are a few limitations to our study. First, since this is a cross-sectional study, the time period between the development of CPSP and acquisition of ^18^F-FDG-PET images cannot be clarified. Second, the specific location other than left/right orientation and characteristics of the pain in the patients were not considered in the analysis of the regions. Third, the sex ratio between the patient and control groups was different; the control group was comprised of men alone, and the patient group included women as well. Finally, this study was limited to the population with pontine hemorrhage, which can be both a strength and limitation. It is now known that a variety of lesions, particularly those of pontine, medullary, thalamic, and cortical strokes can all lead to CPSP [4]. While many previous studies on CPSP have been limited to thalamic lesions or supratentorial strokes or stokes with cortical involvements, this study is meaningful in that it focused on patients with pontine stroke, which is infratentorial stroke [41]. Thus, we could reveal and emphasize that, regardless of stroke lesions, distant areas of the cortex can be involved in CPSP. Further studies, possibly including patients with stroke of the midbrain and medulla, or study comparisons with thalamic strokes can be considered. While previous functional neuroimaging studies of CPSP revealed metabolic changes in the thalamus [42], our results did not indicate metabolic changes in the thalamus. This is significant in that it provides a new perspective that CPSP can occur regardless of any anatomical or functional relationship with the thalamus, and, also, that regions other than the thalamus can be more statistically significantly involved in CPSP.

## 5. Conclusions

In conclusion, our results demonstrated that remote pontine lesions could cause disorientation in multiple regions of the cerebral cortices and cerebellum, which contribute to the experience of pain in CPSP. Future research should be expanded to investigate CPSP in patients of stroke with lesions at different locations, and in association with clinical features using functional neuroimaging studies.

## Figures and Tables

**Figure 1 brainsci-12-00837-f001:**
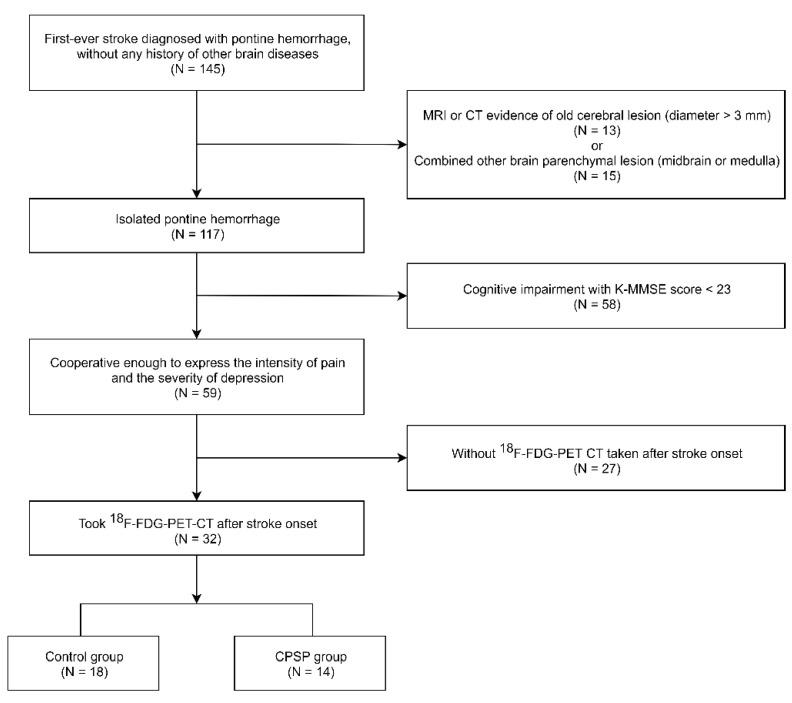
Flowchart of patient enrollment. MRI, magnetic resonance imaging; CT, computed tomography; K-MMSE, Korean Mini-Mental State Examination; ^18^F-FDG-PET, ^18^F-fluorodeoxyglucose-positron emission tomography; CPSP, central post-stroke pain.

**Figure 2 brainsci-12-00837-f002:**
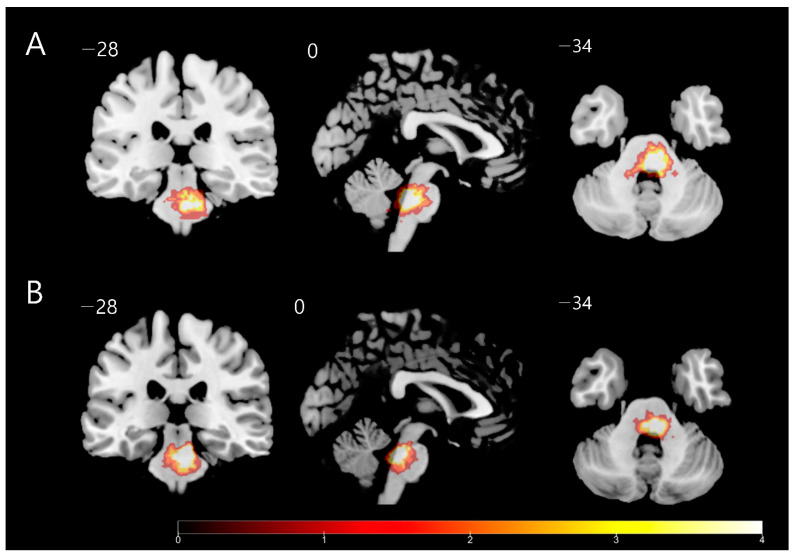
Lesion overlaps in the (**A**) control and (**B**) central post-stroke pain groups in patients with pontine hemorrhage. Left-sided lesions are flipped to the right. Different colors represent numbers of voxels in the superimposed region of interest. Numbers indicate Montreal Neurological Institute z-coordinates.

**Figure 3 brainsci-12-00837-f003:**
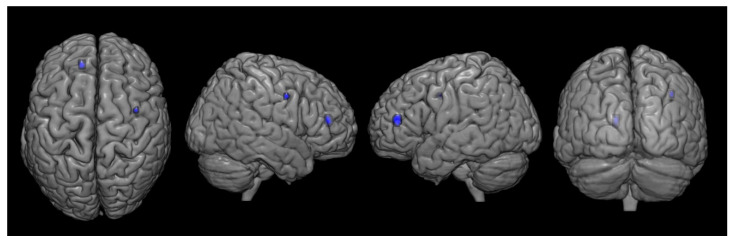
Spatial distributions of decreased glucose metabolism in the central post-stroke pain group following pontine hemorrhage compared with that in the control group. (*P_uncorrected_* < 0.001, k = 10).

**Figure 4 brainsci-12-00837-f004:**
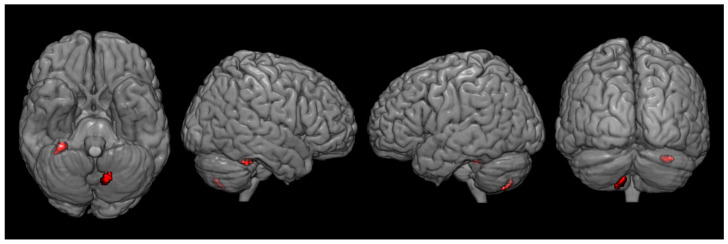
Spatial distributions of increased glucose metabolism in the central post-stroke pain group following pontine hemorrhage compared with that in the control group. (*P_uncorrected_* < 0.001, k = 10).

**Figure 5 brainsci-12-00837-f005:**
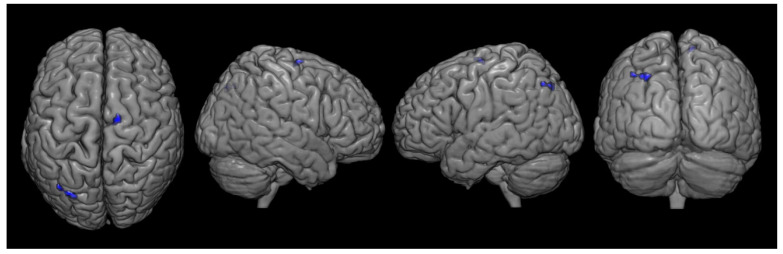
Statistical parametric maps demonstrating the regions with decreased pain intensity correlated with increased pain intensity. (*P_uncorrected_* < 0.001, k = 10).

**Table 1 brainsci-12-00837-t001:** Demographic characteristics of patients with pontine hemorrhage.

	Control Group (n = 18)	CPSP Group (n = 14)	*p*
Age, years	48.1 ± 10.7	50.3 ± 11.6	0.587
Sex (male/female)	18/0	11/3	0.073
Duration since onset n, days	135.2 (123.5)	128.4 (97.3)	0.924
Side of lesion			0.255
Right, 4 (12.5%)	1 (5.6)	3 (21.4)	
Left, 6 (18.8%)	5 (27.8)	1 (7.1)	
Bilateral, 22 (68.7%)	12 (66.7)	10 (71.4)	
Lesion volume, mL	8.4 (5.9)	7.2 (5.7)	0.676
NIHSS (0–42)	9.6 ± 4.9	8.6 ± 4.9	0.586
FM motor (0–100)	50.1 (46.0)	58.7 (51.3)	0.372
FM sensory (0–24)	12.8 (14.3)	11.6 (12.5)	0.746
MMSE (0–30)	28.0 (3.3)	27.4 (4.5)	0.679
GDS (0–30)	12.6 ± 8.4	13.9 ± 9.3	0.665
MQS	7.7 (10.5)	6.6 (8.0)	0.954
Pain intensity (NRS, 0–10)	0 (0.0)	5.6 (1.5)	<0.001 *

Values are presented as means ± standard deviations for normally distributed continuous variables and as medians (interquartile ranges) for non-normally distributed variables. CPSP, central post-stroke pain; NIHSS, National Institutes of Health Stroke Scale; FM, Fugl–Meyer assessment; MMSE, Mini-Mental State Examination; GDS, Geriatric Depression Scale; MQS, Medication Quantification Scale; NRS, numeric rating scale * *p* < 0.05.

**Table 2 brainsci-12-00837-t002:** Location and size of hemorrhage and corresponding pain location and characteristics for each patient with central post-stroke pain.

Patient Number	Location of Hemorrhage	Size of Hemorrhage (mL)	Pain Location	Pain Characteristics	Pain Intensity (NRS)
1	bilateral pons	10.2	Lt. hand	aching, tingling	9
2	bilateral pons	1.0	Rt. whole arm	tingling, cold pain	8
3	bilateral pons	20.3	Lt. whole arm	tingling	7
4	right pons	7.6	Lt. whole arm	aching, tingling	6
5	bilateral pons	10.1	Lt. upper arm, Lt. whole leg	tingling	6
6	left pons	5.4	Rt. face, Rt. whole arm	tingling, stabbing, hyperalgesia	6
7	bilateral pons	8.5	Rt. face, Rt. hand, Rt. foot	tingling	5
8	right pons	2.4	Lt. upper arm, Lt. thigh	aching, tingling	5
9	right pons	3.3	Lt. whole arm, Lt. whole leg	pricking	5
10	bilateral pons	8.9	Lt. buttock	aching	5
11	bilateral pons	7.3	Lt. face, Lt. hand	tingling	5
12	bilateral pons	6.5	Rt. face, Rt. hand	tingling, cold pain	4
13	bilateral pons	5.9	Rt. hand to forearm	pricking, tingling, cold pain	4
14	bilateral pons	3.5	Lt. whole leg	aching, tingling	3

NRS, numeric rating scale.

**Table 3 brainsci-12-00837-t003:** Area of the brain that showed altered glucose metabolism in patients with central post-stroke pain following pontine hemorrhage.

Metabolism	Area	Coordinate			*t* Score	*z* Score	Cluster
		x	y	z			
Decreased	Contralesional Anterior Cingulum	−14	42	14	4.58	3.94	269
	Ipsilesional primary motor cortex	38	0	38	3.84	3.42	224
Increased	Ipsilesional cerebellum	30	−36	−30	4.04	3.57	179
	Contralesional cerebellum	−10	−8	−52	3.92	3.48	292

(*P_uncorrected_* < 0.001, k = 10).

**Table 4 brainsci-12-00837-t004:** Area of the brain that showed altered glucose metabolism in patients with central post-stroke pain showing correlation with intensity of pain.

Metabolism	Area	Coordinate			*t* Score	*z* Score	Cluster
		x	y	z			
Decreased	Ipsilesional Supplementary Motor Cortex	10	−10	66	4.83	3.47	73
	Contralesional angular gyrus	−28	−74	44	4.52	3.33	171

(*P_uncorrected_* < 0.001, k = 10).

## Data Availability

Not applicable.

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
