# Peer review of "Changes in the Brain Metabolism Associated with Central Post-Stroke Pain in Hemorrhagic Pontine Stroke: An 18F-FDG-PET Study of the Brain"

_brainsci, 2022, doi:10.3390/brainsci12070837_

Round 1
Reviewer 1 Report
In the present study the authors analyzed the brain metabolism changes associated with CPSP following pontine hemorrhage. Thirty-two patients with isolated pontine hemorrhage were examined; 14 had CPSP, while 18 did not. Brain glucose metabolism was evaluated using 18F-fluorodeoxyglucose-positron emission tomography images. The authors found that Patients with CPSP showed changes in the brain metabolism in the cerebral cortices and cerebellum. This study emphasizes the role of the many different areas of the cortex that are involved in affective and cognitive processing in the development of CPSP.
The present investigation is the first study to assess the metabolic changes in the brain related to CPSP following isolated brainstem stroke. It provide valuable information to understand the brain functional changes in CPSP involved brain stem hemorrhagic lesion.
The lesion locations in control and CPSP groups were very similar. But the resulted metabolism changes were different in these two groups. Thus it is not clear what are the factors which contribute these differences? One may think of the damages of the somatosensory pathways in the brain stem. But unless the detail structure analysis of the lesion region involved in the brain stem and scrutinize the difference in the lesion region, then there should be other secondary factors contribute these changes and difference.
Although there are several brain regions changes in relation to CPSP. But only decreased metabolism in the ipsilesional supplementary motor area and contralesional angular gyrus was correlated with increased pain intensity, and no region showed positive correlations. Thus it seems that only these two brain areas are relevant to the pain sensitivity. The ACC, primary motor cortex and cerebellum are not directly contributed to the pain sensitivity changes. The authors should add discussion of the functional relation of supplementary motor area and angular gyrus and possible involvement with CPSP.
Author Response
Response to Reviewer 1 Comments
Point 1: In the present study the authors analyzed the brain metabolism changes associated with CPSP following pontine hemorrhage. Thirty-two patients with isolated pontine hemorrhage were examined; 14 had CPSP, while 18 did not. Brain glucose metabolism was evaluated using 18F-fluorodeoxyglucose-positron emission tomography images. The authors found that Patients with CPSP showed changes in the brain metabolism in the cerebral cortices and cerebellum. This study emphasizes the role of the many different areas of the cortex that are involved in affective and cognitive processing in the development of CPSP.
The present investigation is the first study to assess the metabolic changes in the brain related to CPSP following isolated brainstem stroke. It provide valuable information to understand the brain functional changes in CPSP involved brain stem hemorrhagic lesion.
The lesion locations in control and CPSP groups were very similar. But the resulted metabolism changes were different in these two groups. Thus it is not clear what are the factors which contribute these differences? One may think of the damages of the somatosensory pathways in the brain stem. But unless the detail structure analysis of the lesion region involved in the brain stem and scrutinize the difference in the lesion region, then there should be other secondary factors contribute these changes and difference.
Although there are several brain regions changes in relation to CPSP. But only decreased metabolism in the ipsilesional supplementary motor area and contralesional angular gyrus was correlated with increased pain intensity, and no region showed positive correlations. Thus it seems that only these two brain areas are relevant to the pain sensitivity. The ACC, primary motor cortex and cerebellum are not directly contributed to the pain sensitivity changes. The authors should add discussion of the functional relation of supplementary motor area and angular gyrus and possible involvement with CPSP.
Response 1: We thank the reviewer for pointing this out. In the discussion (3rd paragraph) on page 7-8, the association of supplementary motor cortex and angular gyrus with the emotional aspects of pain and with integrative role in sensory inputs are described. We have added additional relationship of these regions with pain, and made it as a separate paragraph to come before other paragraphs. We highly appreciate the reviewer’s insightful comments on improving the readability of the paper.
Accordingly, the manuscript was revised as follows:
The supplementary motor area and angular gyrus were the only regions that appeared to be relevant to the pain sensitivity. Many aspects of the supplementary motor cortex remain questionable; however, its relation to emotion, affective functions, and cognitive control and behavior processing are known. Moreover, its functional connections with the limbic system and primary motor cortex have been proven to play a role in negative emotions. A study of rheumatoid arthritis patients using functional MRI (fMRI) reported increased connectivity predominately for the supplementary motor cortex, cingulate cortex and bilateral sensorimotor cortex, suggesting their involvement in pain processing. The angular gyrus is believed to play an integrative role; multisensory inputs are integrated in the angular gyrus, and interactions with different subsystems such as memory, attention, and concepts are performed to ultimately comprehend events, thus acting as an attentional subsystem. The angular gyrus thus reflects the ability to integrate aspects of information, especially including sensory information and internal mental representations.

Reviewer 2 Report
Choi and colleagues investigated the association between CPSP and changes in brain metabolism in hemorrhagic pontine stroke. The results provide some insight into this field. However, there are some drawbacks in the study that have affected the quality of this manuscript.
1. The authors set the uncorrected p value < .001. It appears that the authors did not adjust for multiple comparisons. Considering the small sample size of the study, this is a major limitation regarding the validity of the study results. It is likely this study is under power and thus, the significant findings could be inflated. Similarly, it is unclear whether authors adjusted for multiple regression analyses. Please report the results after adjusting multiple comparisons.
2. In Discussion (the 3rd paragraph), the authors mentioned that the ACC was associated with CPSP in the study. This was not reported in the Results. Please add this information to the Results or remove the text regarding ACC in Discussion.
3. In Discussion (4th paragraph), the authors talked about M1 in chronic pain. However, it seems to be out of place as M1 was not mentioned in the Introduction or Results and its link to the brain metabolic changes in the study was not stated. Please establish a clear link of M1 to findings of this study.
4. The 5th paragraph of Discussion focuses on cerebellum in chronic pain. However, this study did not find an association between CPSP and cerebellum. Please provide an explanation on the insignificant findings on relationship between cerebellum and CPSP.
5. Could the authors elaborate why the study focused on pontine stroke is a limitation as this was justified in your Introduction?
6. Please elaborate the finding that this study did not find any abnormality in the thalamus, comparing with previous studies.
7. In Conclusion, as this is a cross-sectional study, any inference on causal relationship is misleading. Please remove text in line 252-256 on page 8.
Author Response
Response to Reviewer 2 Comments
Point 1: The authors set the uncorrected p value < .001. It appears that the authors did not adjust for multiple comparisons. Considering the small sample size of the study, this is a major limitation regarding the validity of the study results. It is likely this study is under power and thus, the significant findings could be inflated. Similarly, it is unclear whether authors adjusted for multiple regression analyses. Please report the results after adjusting multiple comparisons.
Response 1: We thank the reviewer for pointing this out. Voxel wise corrections for multiple comparisons implemented in neuroimaging packages, such as SPM, have been shown to be conservative, repeatedly. Owing to the limited variability of the lesions, we expected to find small effects within brain cortices. Therefore, we decided to identify changes at a statistical voxel wise uncorrected threshold of two-tailed P < .001. To reduce false-positive findings, a cluster size threshold of 10 continuous voxels were required for significance, and we later replicated the results with a regions-of-interest approach using volumetric measurements. We have added this explanation in Materials and Methods (2.4. Analyses of brain 18F-FDG-PET images, last paragraph).
Accordingly, the manuscript was revised as follows:
Voxel wise corrections for multiple comparisons implemented in neuroimaging packages, such as SPM, have been shown to be conservative, repeatedly. Owing to the limited variability of the lesions, we expected to find small effects within brain cortices. Therefore, we decided to identify changes at a statistical voxel wise uncorrected threshold of two-tailed P < .001. To reduce false-positive findings, a cluster size threshold of 10 continuous voxels were required for significance, and we later replicated the results with a regions-of-interest approach using volumetric measurements.
Point 2: In Discussion (the 3rd paragraph), the authors mentioned that the ACC was associated with CPSP in the study. This was not reported in the Results. Please add this information to the Results or remove the text regarding ACC in Discussion.
Response 2: Thank you for pointing this out. In the results (2nd paragraph) on page 5, it is mentioned that compared with the control group, the CPSP group showed significant hypometabolism in the contralesional rostral anterior cingulum. We have changed the word anterior cingulum to anterior cingulate cortex in the results for the consistency of the manuscript and to reduce confusion for the readers.
Point 3: In Discussion (4th paragraph), the authors talked about M1 in chronic pain. However, it seems to be out of place as M1 was not mentioned in the Introduction or Results and its link to the brain metabolic changes in the study was not stated. Please establish a clear link of M1 to findings of this study.
Response 3: We thank the reviewer for pointing this out. In the results (2nd paragraph) on page 5, it is stated that the CPSP group showed hypometabolism in the ipsilesional primary motor cortex, which is M1. We have added M1 in parentheses behind primary motor cortex in the result, to reduce confusion. We highly appreciate the reviewer’s insightful comments on improving the readability of the paper.
Point 4: The 5th paragraph of Discussion focuses on cerebellum in chronic pain. However, this study did not find an association between CPSP and cerebellum. Please provide an explanation on the insignificant findings on relationship between cerebellum and CPSP.
Response 4: Thank you for pointing this out. As mentioned in the results (2nd paragraph) on page 5, in comparison to the control group, the CPSP patients showed increased brain metabolism in the ipsilesional cerebellum (lobule VI) and contralesional cerebellum (lobule VIIB), which are both in the superior posterior lobe of the cerebellum. Since cerebellar cognitive affective syndrome is known to result from injury to the posterior cerebellar lobe, this makes the posterior lobe critical for cognitive and affective processing. Therefore, this study suggests that cognitive processing in areas of cerebellum may be related to nociceptive processing and pain perception.
Point 5: Could the authors elaborate why the study focused on pontine stroke is a limitation as this was justified in your Introduction?
Response 5: Thank you for your suggestion. It is now known that a variety of lesions, particularly those of pontine, medullary, thalamic and cortical strokes can all lead to CPSP. This study is meaningful in that it focused on patients with pontine stroke, which is infratentorial stroke, whereas many previous studies on CPSP have been limited to thalamic lesions or supratentorial strokes or stokes with cortical involvements. Thus, we could reveal and emphasize that regardless of stroke lesions, distant areas of the cortex can be involved in CPSP. However, for the quality of 18F-FDG-PET study regarding structural and lesional homogeneity of the enrolled population, we focused on pontine stroke among infratentorial strokes. Broader studies including other infratentorial strokes, such as midbrain and medullary strokes, or population expansion to include ischemic strokes will be meaningful and may reveal other functionally connected regions other than the results of our study. We have added more detailed explanation to the limitation in Discussion (last paragraph).
Accordingly, the manuscript was revised as follows:
It is now known that a variety of lesions, particularly those of pontine, medullary, thalamic and cortical strokes can all lead to CPSP. While many previous studies on CPSP have been limited to thalamic lesions or supratentorial strokes or stokes with cortical involvements, this study is meaningful in that it focused on patients with pontine stroke, which is infratentorial stroke. Thus, we could reveal and emphasize that regardless of stroke lesions, distant areas of the cortex can be involved in CPSP. Further studies, possibly including patients with stroke of the midbrain and medulla, or study comparisons with thalamic strokes can be considered.
Point 6: Please elaborate the finding that this study did not find any abnormality in the thalamus, comparing with previous studies.
Response 6: We thank the reviewer for pointing this out. According to the designation of the P value threshold (0.001), only the regions with the greatest correlation was left. The results of this study therefore means that the areas of the results are statistically more significantly involved in CPSP than thalamus. This might be because previous studies mostly included patients with structural damage of the thalamus or with supratentorial lesions, while the population in our study did not. This is significant in that it provides a new perspective that CPSP can occur regardless of any anatomical or functional relationship with the thalamus, and also that regions other than the thalamus can be more statistically significantly involved in CPSP. We have added additional explanation in the Discussion (last paragraph).
Point 7: In Conclusion, as this is a cross-sectional study, any inference on causal relationship is misleading. Please remove text in line 252-256 on page 8.
Response 7: We agree with this and have incorporated your suggestion throughout the manuscript. We have removed text in line 252-256 in Conclusion.
